# *Gynostemma Pentaphyllum* Extract Ameliorates High-Fat Diet-Induced Obesity in C57BL/6N Mice by Upregulating SIRT1

**DOI:** 10.3390/nu11102475

**Published:** 2019-10-15

**Authors:** Hyun Sook Lee, Su-Min Lim, Jae In Jung, So Mi Kim, Jae Kyoung Lee, Yoon Hee Kim, Kyu Min Cha, Tae Kyu Oh, Joo Myung Moon, Tae Young Kim, Eun Ji Kim

**Affiliations:** 1Department of Food Science & Nutrition, Dongseo University, Busan 47011, Korea; hyunlee@dongseo.ac.kr; 2Regional Strategic Industry Innovation Center, Hallym University, Chuncheon 24252, Korea; sumin8481@hallym.ac.kr (S.-M.L.); jungahoo@hallym.ac.kr (J.I.J.); somisss@hallym.ac.kr (S.M.K.); 3Technology Development Center, BTC Corporation, Ansan 15588, Korea; ljk@btcbio.com (J.K.L.); kyh@btcbio.com (Y.H.K.); ckm@btcbio.com (K.M.C.); otk@btcbio.com (T.K.O.); mhjj1919@btcbio.com (J.M.M.); tykim@btcbio.com (T.Y.K.)

**Keywords:** *Gynostemma pentaphyllum*, obesity, gypenoside, ginsenoside, AMPK, SIRT1

## Abstract

*Gynostemma pentaphyllum* is widely used in Asia as a herbal medicine to treat type 2 diabetes, dyslipidemia, and inflammation. Here, we investigated the anti-obesity effect and underlying mechanism of *G. pentaphyllum* extract (GPE) enriched in gypenoside L, gypenoside LI, and ginsenoside Rg3 and obtained using a novel extraction method. Five-week-old male C57BL/6N mice were fed a control diet (CD), high-fat diet (HFD), HFD + 100 mg/kg body weight (BW)/day GPE (GPE 100), HFD + 300 mg/kg BW/day GPE (GPE 300), or HFD + 30 mg/kg BW/day Orlistat (Orlistat 30) for 8 weeks. The HFD-fed mice showed significant increases in body weight, fat mass, white adipose tissue, and adipocyte hypertrophy compared to the CD group; but GPE inhibited those increases. GPE reduced serum levels of triglyceride, total cholesterol, and LDL-cholesterol, without affecting HDL-cholesterol. GPE significantly increased AMPK activation and suppressed adipogenesis by decreasing the mRNA expression of CCAAT/enhancer binding protein-α (C/EBPα), peroxisome proliferator-activated receptor-γ (PPARγ), sterol regulatory element-binding protein-1c (SREBP1c), PPARγ coactivator-1α, fatty acid synthase (FAS), adipocyte protein 2 (AP2), and sirtuin 1 (SIRT1) and by increasing that of carnitine palmitoyltransferase (CPT1) and hormone- sensitive lipase (HSL). This study demonstrated the ameliorative effect of GPE on obesity and elucidated the underlying molecular mechanism.

## 1. Introduction

The prevalence of obesity has increased worldwide over the past 50 years and has reached pandemic levels. Once considered a problem only in high-income countries, obesity is now dramatically on the rise in low- and middle-income countries. Obesity is a major life-threatening health problem because it substantially increases the risk of various chronic diseases, such as diabetes, cardiovascular diseases, and cancers, thus contributing to a decline in both quality of life and life expectancy [1,2,3]. Therefore, the prevention and treatment of obesity are extremely important. Several anti-obesity drugs have been marketed in the last few years but have been successively withdrawn because of serious adverse effects. Only a few drugs are currently prescribed for treating obesity [4,5,6]. Herbal remedies are recognized as an important component of human health care as they have few side effects [7]. Therefore, there is a growing interest in herbal remedies and functional foods [4,5,6]. However, low bioavailability is a concern, as it may limit or even interfere with the effectiveness of herbal remedies and remains a major challenge in the development of clinically useful functional foods and medicines. The development of appropriate extraction methods to increase bioavailability and increase the contents of effective compounds may offer a solution.

*Gynostemma pentaphyllum* (Thunb.) Makino is an herbaceous climbing vine of the family Cucurbitaceae that is widely distributed in South and East Asia. Its leaves have been traditionally used in Korea, China, and Japan as a herbal medicine or tea. *G. pentaphyllum* is also widely used as a health supplement in beverages, biscuits, noodles, face washes, and bath oils [8]. Because of its health benefits, *G. pentaphyllum* has recently attracted increasing attention. Phytochemical studies have revealed that *G. pentaphyllum* contains saponins, flavonoids, polysaccharides, amino acids, vitamins, and some essential elements [9,10,11,12]. In vitro and in vivo studies have suggested that extracts of *G. pentaphyllum* exert various beneficial bioactivities, including antimicrobial [13], antioxidant [14,15], anticancer [16,17], anti-inflammatory [17,18], antidiabetic [19,20,21], antilipidemic [22], neuroprotective [23], and anti-obesity [24,25] activities. The major bioactive phytochemicals of *G. pentaphyllum* are dammarane-type triterpene saponins called gypenosides [26,27]. Around 180 gypenosides have been identified in *G. pentaphyllum* to date [28] and have been reported to have numerous beneficial activities, such as antioxidant [29], anticancer [30], hypoglycemic [31], hypolipidemic [32], and hepatoprotective [29,32] activities. These beneficial activities are related to the levels and characteristics of the gypenosides, including molecular weight, monosaccharide composition, and chemical structure, which are influenced by the extraction and purification methods used to obtain specific gypenoside-enriched products [33]. Therefore, to produce and apply *G. pentaphyllum* extract to functional foods and nutraceuticals, standardization to control the amount and characteristics of gypenosides is required.

In Vivo studies have indicated that *G. pentaphyllum* is a safe substance that does not cause side effects, even when ingested over a long period. No toxicity or mortality was observed upon long-term administration of up to 750 mg/kg *G. pentaphyllum* in rats [34]. Standardized water extract of *G. pentaphyllum* showed no toxicity in rats [35]. Therefore, *G. pentaphyllum* is expected to be safe for use as a chemotherapeutic agent. Recently, we developed a *G. pentaphyllum* extract (GPE) with much higher contents of gypenoside L (1.8%, w/w), gypenoside LI (1.4%, w/w), and ginsenoside Rg3 (0.15%, w/w) than extracts produced by conventional methods [36]. In an in vitro study using L6 skeletal muscle cells, we found that GPE stimulated glucose uptake via activation of AMP-activated protein kinase (AMPK) [36]. The current study aimed to evaluate the in vivo anti-obesity efficacy of GPE. To this end, dietary obesity was induced in C57BL/6N mice by feeding with a high-fat diet (HFD). The effects of GPE at various concentrations on body weight, fat mass, blood lipid profile, and adipogenic transcription factors and their target genes were examined.

## 2. Materials and Methods

### 2.1. Preparation of GPE

GPE was prepared at BTC Corporation according to a previously described method [36]. In brief, the leaves of *G. pentaphyllum* were dried using an electric heat controller (i.e., electric fan). Extracts were prepared from 1 kg of leaves with hot water (20 L) and 50% EtOH aqueous solution (15 L). Supernatants of the two extractions were collected and combined, and then filtered. The filtrate was vacuum-evaporated to obtain GPE. Gypenoside L, gypenoside LI, and gynsenoside Rg3 were quantified by high-performance liquid chromatography. GPE contains gypenoside L (1.8%, w/w), gypenoside LI (1.4%, w/w), and ginsenoside Rg3 (0.15%, w/w).

### 2.2. Animals

Four-week-old male C57BL/6N mice were purchased from Doo Yeol Biotech Co. Ltd. (Seoul, Korea) and housed in controlled standard conditions (23 ± 3 °C, 50 ± 10% relative humidity, and a 12-h light/dark cycle). Mice were acclimatized for one week before use and had free access to a standard non-purified rodent diet (Cargill Agri Purina, Inc., Seongnam, Korea) and water. All animal experiments were conducted according to the protocols ratified by the Institutional Animal Care and Use Committee of Hallym University (approval number: Hallym 2018-31).

### 2.3. Experimental Design and Treatment

After one week of acclimation, C57BL/6N mice were randomly divided into five groups (*n* = 10 per group) fed the following diets: (1) control diet (CD); (2) HFD; (3) HFD + 100 mg/kg body weight (BW)/day GPE (GPE 100); (4) HFD + 300 mg/kg BW/day GPE (GPE 300); (5) HFD + 30 mg/kg BW/day Orlistat (positive control material) (Orlistat 30). The CD (containing 10 kcal% as fat, Cat No. D12450B) and HFD (containing kcal% as fat, Cat No. D12452) were purchased from Research Diets, Inc. (New Brunswick, NJ, USA). The mice were given feed and water ad libitum throughout the study period. GPE or Orlistat dissolved in saline solution was administered daily by oral gavage for eight weeks. Mice in the CD and HFD groups were given an equal volume of saline solution by oral gavage at the same time. Body weight was measured once a week and food intake monitored daily throughout the experimental period.

At the end of the experiment, all mice were fasted for 16 h and anesthetized with tribromoethanol diluted in tertiary amyl alcohol. Whole body composition (lean body mass percentage and fat mass percentage) was measured using dual-energy X-ray absorptiometry (PIXImus^TM^; GE Lunar, Madison, WI, USA). Then, blood was collected from the orbital vein and serum was obtained by centrifugation at 3,000 rpm at 4 °C for 20 min. Adipose tissues were rapidly excised, rinsed, and weighed.

### 2.4. Biochemical Analyses of Serum

Serum levels of glucose, triglyceride, total cholesterol, low-density lipoprotein (LDL)-cholesterol, and high-density lipoprotein (HDL)-cholesterol, as well as aspartate aminotransferase (AST) and alanine aminotransferase (ALT) activities were measured using a blood chemistry autoanalyzer (KoneLab 20XT, Thermo Fisher Scientific, Vantaa, Finland).

### 2.5. Histological Analysis

Epididymal adipose tissues were fixed in 4% paraformaldehyde solution, embedded in paraffin, and cut into 5-μm sections. Tissue sections were stained with hematoxylin and eosin (H&E) and examined and imaged under a light microscope (AxioImager; Carl Zeiss, Jena, Germany). Adipocyte number and size were quantified using the AxioVision Imaging System (Carl Zeiss) with a magnifying power of 200×. Slides were examined in a blinded manner.

### 2.6. Quantitative Reverse Transcription PCR (RT-qPCR)

Total RNA was extracted from the epididymal adipose tissues with Trizol (Cat No. 15596018, Invitrogen Life Technologies, Carlsbad, CA, USA) according to the manufacturer’s instructions. RNA content and purity were measured using a micro-volume spectrophotometer (BioSpec-nano, Shimadzu, Kyoto, Japan). RT-qPCR was conducted using a Rotor-Gene 3000 instrument (Corbett Research, Mortlake, Australia) and Rotor-Gene^TM^ SYBR Green kit (Cat No. 204074, Qiagen, Valencia, CA, USA) according to the manufacturer’s instruction. The sequences of the primers used in this study are listed in Table 1. The primers used in this study were ordered and synthesized in Bioneer Co. (Daejeon, Korea). Thermal cycles were: 94 °C for 3 minutes, 40 cycles of 95 °C for 10 seconds, 60 °C for 15 seconds, and 72 °C for 20 seconds. The results were analyzed with Rotor-Gene 6000 Series System Software program, version 6 (Corbett Research), and target mRNA levels were normalized to that of glyceraldehyde 3-phosphate dehydrogenase (*Gapdh*).

### 2.7. Western Blotting Analysis

Epididymal adipose tissue samples were homogenized with a polytron tissue homogenizer in ice-cold lysis buffer (20 mM HEPES, pH 7.5, 150 mM NaCl, 1 mM EDTA, 1 mM EGTA, 100 mM NaF, 10 mM sodium pyrophosphate, and 1% Triton X-100) containing 5 mM iodoacetic acid and protease inhibitor (iNtRON Biotechnology, Inc., Seongnam, Korea), then solubilized at 4 °C for 40 min. Insoluble material was removed by centrifugation at 14,800× *g* for 10 min and the supernatant was collected and used for Western blot analyses. The protein contents of the lysates were measured using a BCA protein assay kit (Thermo Scientific, Rockford, IL, USA). Western blot analyses were conducted as described previously [37]. The antibodies against p-AMPKα (T172) (Cat No. 2535), AMPKα (Cat No. 2532), and β-actin (Cat No. 3700) were obtained from Cell Signal Technology (Beverly, MA, USA). Anti-rabbit IgG HRP-linked antibody (Cat No. 7074) and anti-mouse IgG HRP-linked antibody (7076) were obtained from Cell Signal Technology. Blots were detected with Luminata^TM^ Forte Western HRP Substrate (Cat No. WBLUF0500, Millipore, Billerica, MA, USA). The relative protein abundances were determined using the ImageQuant^TM^ LAS 500 imaging system (GE Healthcare Bio-Sciences AB, Uppsala, Sweden). Protein expression levels were normalized to that of β-actin.

### 2.8. Statistical Analysis

All data are expressed as the mean ± SD. Student’s *t*-test was used to test differences between the CD and HFD groups. Analysis of variance (ANOVA) followed by Duncan’s multiple comparison test was used to compare means between the HFD, GPE 100, GPE 300, and Orlistat 30 groups. *P* < 0.05 was considered significant.

## 3. Results

### 3.1. Effects of GPE on Body Weight and Fat Mass in HFD-Induced Obese C57BL/6N Mice

Consumption of HFD for eight weeks caused significant increases in body weight (Table 2), but mean body weight gain and fat mass percentage were significantly lower in the GPE-treated groups than in the HFD group. Orlistat, a commonly used weight loss agent, has a positive effect on HFD-induced obesity in murine models [38]. Notably, a greater anti-obesity effect (in terms of weight gain) was observed in the GPE 300 group than in the Orlistat 30 group. Lean body mass was significantly decreased by HFD (*P* < 0.001), but this decrease tended to be suppressed by GPE treatment. Food intake was significantly lower in the HFD group than in the CD group (*P* < 0.001). The GPE 100 and GPE 300 groups had significantly lower food intake than the HFD group. The food efficiency ratio was significantly higher in the HFD group than in the CD group and was lowered dose-dependently by GPE treatment (*P* < 0.001).

### 3.2. Effects of GPE on Serum Glucose, Lipid, ALT, and AST Levels

Serum glucose was significantly increased in the HFD group compared to the CD group (*P* < 0.001). Treatment with GPE dose-dependently attenuated the increase in serum glucose in HFD-fed mice (Table 3). Serum triglyceride, total cholesterol, and LDL-cholesterol levels were also significantly increased by HFD feeding (*P* < 0.001, *P* < 0.01, *P* < 0.01, respectively). Serum triglyceride was significantly reduced in both the GPE 100 and GPE 300 groups compared to the HFD group. Serum total cholesterol was decreased in the GPE 300 group compared to the HFD group. LDL-cholesterol level was significantly decreased by GPE treatment. On the other hand, serum HDL-cholesterol was significantly increased in the HFD group compared to the CD group (*P* < 0.05), and GPE or Orlistat treatment had no effect on serum HDL-cholesterol (Table 3). ALT and AST activity levels were measured to determine liver toxicity by GPE treatment. ALT and AST activities were significantly increased by HFD feeding (*P* < 0.001, *P* < 0.01, respectively), and the increases were significantly suppressed by GPE treatment (Table 3).

### 3.3. Effects of GPE on Adipose Tissue Accumulation

To investigate the effect of GPE on adipose tissue weight, the weight of white adipose tissue (epididymal, retroperitoneal, mesenteric, and inguinal fat) was monitored. As shown in Table 4, all white adipose tissues were significantly increased (albeit to different levels) in the HFD group compared to the CD group (*P* < 0.001). When GPE was given, the increase in weight of each adipose was dose-dependently suppressed. Figure 1A shows that the adipocyte size in epididymal adipose tissues was larger in the HFD group than in the CD group, and GPE treatment suppressed this adipocyte enlargement. The number of adipocytes was significantly higher in the HFD group than in the CD group (*P* < 0.001). GPE treatment significantly lowered the number of epididymal adipocytes with a diameter >120 µm (Figure 1B).

### 3.4. Effects of GPE on the Expression of Key Factors in Adipogenesis and Fat Oxidation

To investigate the molecular basis for the anti-obesity effect of GPE, we measured the expression of adipogenic transcription factors, including CCAAT/enhancer binding proteinα (C/EBPα), peroxisome proliferator-activated receptor-γ (PPARγ), and sterol regulatory element-binding protein-1c (SREBP-1c) as well as their target genes in epididymal adipose tissues. Compared to the CD group, the HFD group exhibited dramatically increased *Cebpa*, *Pparg*, and *Srebp1c* mRNA expression (*P* < 0.001). GPE treatment noticeably suppressed the elevated mRNA expression of these three transcription factors by HFD (Figure 2). As for fatty acid biosynthesis, the mRNA levels of fatty acid synthase (*Fas*) and adipocyte protein 2 (*Ap2*) were higher in the HFD group than in the CD group, whereas they were significantly reduced in the GPE 100 and GPE 300 groups compared to the HFD group. As for fatty acid oxidation, carnitine palmitoyltransferase 1 (*Cpt1*) mRNA expression was higher in the HFD group than in the CD group. *Cpt1* mRNA expression in the GPE 100 group was not different from that in the HFD group, but it was significantly increased in the GPE 300 group when compared with the HFD and GPE 100 groups. Hormone sensitive lipase (*Hsl*) mRNA expression was significantly decreased in the HFD group compared with the CD group but was dose- dependently increased by GPE treatment (Figure 3).

### 3.5. Effects of GPE on SIRT1 Expression and AMPK Activation

Sirtuin 1 (SIRT1), an epigenetic enzyme involved in protein deacetylation, is the most conserved mammalian NAD^+^-dependent protein deacetylase. It acts primarily as a metabolic sensor that responds to changing energy status by deacetylating crucial transcription factors and cofactors. SIRT1 regulates diverse biological functions, including biogenesis, inflammation, apoptosis, oxidative stress, mitochondrial function, and cellular senescence [39]. We analyzed *Sirt1* expression to determine whether the changes in expression of adipogenic transcription factors and their target genes by GPE treatment were regulated by SIRT1. As shown in Figure 4, SIRT1 mRNA expression was significantly increased in the HFD group compared to the CD group (*P* < 0.05), and in the GPE100 and GPE300 groups compared to the HFD group (*P* < 0.05).

AMPK protein expression was significantly decreased in the HFD group compared to the CD group and was not affected by GPE treatment. However, AMPK phosphorylation (p-AMPK) was significantly reduced in the HFD group compared to the CD group, whereas GPE treatment increased the p-AMPK levels (Figure 5A). Thus, the p-AMPK/AMPK ratio was decreased by HFD feeding, but dose-dependently increased by GPE treatment (Figure 5B).

## 4. Discussion

The present study demonstrated that GPE treatment reduced body weight, body weight gain, fat mass, white adipose tissue weight, food intake, and food efficiency ratio in C57BL/6N mice with HFD-induced obesity. These results clearly showed that GPE has a potent anti-obesity effect. 

In our study, blood glucose was increased by HFD feeding, which was dramatically reduced by GPE treatment. These hypoglycemic effects of *G. pentaphyllum* have been reported in other studies [24,40,41]. Wang et al. reported that polysaccharides extracted from *G. pentaphyllum* decreased α-glucosidase activity and increased GLUT2 protein expression in diabetic mice [41]. Gao et al. reported that *G. pentaphyllum* exerted hypoglycemic and hypolipidemic effects through NFE2-related factor 2 signaling in streptozotocin-induced diabetic rats [31]. Gauhar et al. found that a *G. pentaphyllum* extract named actiponin obtained by a modified method combining autoclaving and ethanol extraction contained large amounts of damulin A and damulin B and induced a hypoglycemic effect by increasing AMPK and ACC phosphorylation [24]. In our study, the increases in the serum levels of triglyceride, total cholesterol, and LDL-cholesterol induced by HFD also were remarkably suppressed by GPE treatment. *G. pentaphyllum* has been reported to improve blood lipid levels in Zucker fatty rats [42] and obese humans [25]. Further, our study showed that the increases in serum ALT and AST activities induced by HFD feeding were significantly suppressed by GPE treatment. These observations indicate that our GPE, which is enriched in gypenoside L, gypenoside LI, and ginsenoside Rg3, ameliorates hyperglycemic and hyperlipidemic symptoms without being toxic in HFD-induced obese C57BL/6N mice.

Adipogenesis-related transcription factors such as C/EBPα, PPARγ, and SREBP-1c act as key regulators of glucose and energy metabolism as well as of preadipocyte differentiation and lipid biosynthesis [43,44,45]. In our study, GPE treatment noticeably suppressed the increases in the mRNA expression of *Cebpa*, *Pparg*, and *Srebp1c* induced by HFD feeding. In the present study, GPE treatment suppressed HFD-induced increases in the mRNA expression of *Fas* and *Ap2*, involved in fatty acid biosynthesis, but increased the HFD-induced reduction in the mRNA expression of *Cpt1* and *Hsl*, involved in fatty acid oxidation and lipolysis, in adipose tissue. FAS is a key enzyme in de novo lipogenesis, and, when activated, it increases fatty acid synthesis and insulin resistance in adipose tissues [46]. aP2, also called fatty acid binding protein 4, is used as an early biomarker of metabolic syndrome. Blocking this protein may provide a treatment for various chronic diseases, including heart disease, diabetes, and obesity [47,48,49]. CPT1 is an essential enzyme in fatty acid β-oxidation and is associated with diabetes and obesity [50]. HSL is an intracellular neutral lipase capable of hydrolyzing a variety of esters. In adipose tissue, HSL hydrolyzes stored triglycerides to free fatty acids to mobilize stored fats [51]. Taken together, our results indicate that the downregulation of transcription factors such as C/EBPα, PPARγ, and SREBP-1c, followed by the downregulation of FAS and aP2 and the upregulation of CPT1 and HSL, resulting from GPE treatment are part of the mechanism by which GPE exhibits anti-obesity effects.

AMPK is an important intracellular sensor that controls the energy balance in the whole body and cells by phosphorylating key target proteins related to lipid metabolism, fatty acid oxidation, and glucose uptake in response to energy supply and demand. When AMPK is activated, the ATP-generating pathway is promoted through increases in fatty acid and glucose oxidation, whereas the ATP-consuming pathway is suppressed through decreases in lipid and cholesterol synthesis [52,53,54]. Owing to its effect on glucose and lipid metabolism, AMPK activation has been studied for the treatment of metabolic syndrome [25,54,55]. Metformin [56,57], 5-aminoimidazole-4-carboxamide-1-β-d-ribofuranoside [58,59], A-769662 (a thienopyridone AMPK activator) [55], and damulin [25] reportedly improve diabetes, dyslipidemia, and fatty liver. The commonality of these substances is that they activate AMPK. Total extract or total saponins of *G. pentaphyllum* have been reported to have diverse effects downstream of AMPK activation [24]. In our previous in vitro study, GPE and gypenoside L, gypenoside LI, and ginsenoside Rg3 were found to activate AMPK [36]. The present study confirmed that GPE treatment activates AMPK in HFD-induced obese C57BL/6N mice. Among the components of *G. pentaphyllum* and damulin A and B are major AMPK activators [24,25,60]. Our study showed that gypenoside L-, gypenoside LI-, and ginsenoside Rg3-enriched GPE also acts as an effective AMPK activator.

SIRT1 is an NAD^+^-dependent protein deacetylase that has emerged as a key metabolic regulator in various metabolic tissues. SIRT1 couples the deacetylation of various transcription factors and co- factors to the cleavage of NAD^+^, an indicator of cellular metabolic status, playing a vital role in metabolism, including fat storage, gluconeogenesis, fatty acid oxidation, lipogenesis, insulin secretion, food intake, circadian rhythm, and inflammation [40]. SIRT1 promotes fat mobilization in white adipocytes by repressing PPARγ, thereby suppressing the expression of adipose tissue markers, such as aP2 [61]. Genetic ablation of *Sirt1* in adipose tissues leads to increased adiposity and insulin resistance [62]. Resveratrol, which activates SIRT1, inhibits HFD-induced obesity. AMPK activation is required for SIRT1 action [63], and AMPK regulates energy expenditure through the regulation of NAD^+^ metabolism and SIRT1 activity [64]. We analyzed *Sirt1* mRNA expression in the epididymal adipose tissue to determine whether the anti-obesity effect of GPE was due to a change in SIRT1 expression. We found that *Sirt1* was significantly increased in the HFD group compared to the CD group and was dose-dependently increased by GPE treatment.

## 5. Conclusions

In conclusion, the GPE produced in this study, enriched in gypenoside L, gypenoside LI, and ginsenoside Rg3, has potent anti-obesity activity. Our results suggest that the underlying mechanism of action involves AMPK phosphorylation, which leads to increased SIRT1 expression, which in turn reduces PPARγ expression, ultimately affecting targets of PPARγ, including FAS, aP2, HSL, and CPT1. Therefore, GPE is a potential candidate for the development of therapies for obesity and related diseases.

## Figures and Tables

**Figure 1 nutrients-11-02475-f001:**
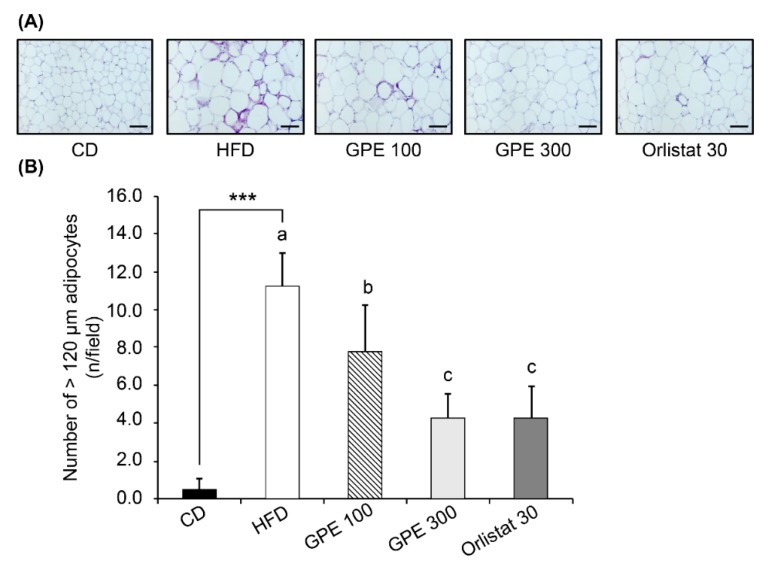
Effect of GPE treatment on morphological changes in epididymal adipose tissues in HFD-fed C57BL/6N mice. GPE was administrated by oral gavage for 8 weeks to mice fed an HFD. The epididymal adipose tissue was excised, fixed with 4% paraformaldehyde, embedded in paraffin, and cut into 5-μm sections. Tissue sections were stained with H&E. Adipocyte number and size were quantified. (**A**) Representative H&E-stained epididymal adipose tissue sections (*n* = 10). 200× magnification, Scale bar = 100 μm. (**B**) The number of adipocytes over 120 μm was counted. Each bar represents the mean ± SD (*n* = 10). *** *P* < 0.001 significantly different from the CD group. Different letters indicate significant differences among the HFD, GPE 100, GPE 300, and Orlistat 30 groups at *P* < 0.05. CD: control diet; HFD: high-fat diet; GPE: *G. pentaphyllum* extract; GPE 100: HFD + 100 mg/kg body weight/day; GPE 300: HFD + 300 mg/kg body weight/day GPE; Orlistat 30: HFD + 30 mg/kg body weight/day Orlistat.

**Figure 2 nutrients-11-02475-f002:**
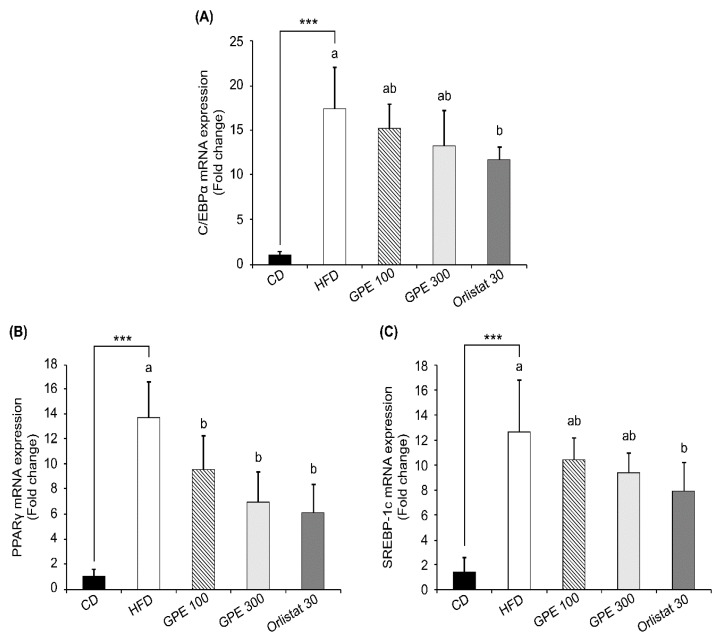
Effect of GPE treatment on the expression of adipogenic transcription factors in epididymal adipose tissues of HFD-fed C57BL/6N mice. GPE was administrated by oral gavage for 8 weeks to mice fed an HFD. Total RNA was isolated from epididymal adipose tissues, reverse transcribed, and used for qRT-PCR. Relative mRNA expression levels of *Cebpa* (**A**), *Pparg* (**B**), and *Srebp1c* (**C**) in adipose tissues. Target mRNA expression was normalized to that of *Gapdh* and is presented relative to the CD group. Each bar represents the mean ± SD (*n* = 10). *** *P* < 0.001 significantly different from the CD group. Different letters indicate significant differences among the HFD, GPE 100, GPE 300, and Orlistat 30 groups at *P* < 0.05. CD: control diet; HFD: high-fat diet; GPE: *G. pentaphyllum* extract; GPE 100: HFD + 100 mg/kg body weight/day; GPE 300: HFD + 300 mg/kg body weight/day GPE; Orlistat 30: HFD + 30 mg/kg body weight/day Orlistat.

**Figure 3 nutrients-11-02475-f003:**
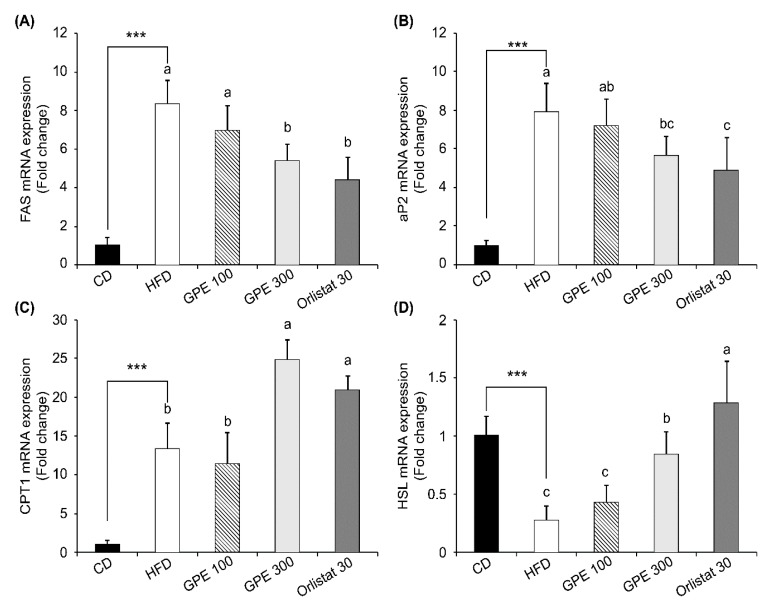
Effect of GPE treatment on the expression of adipogenesis- and lipolysis-related genes in epididymal adipose tissues of HFD-fed C57BL/6N mice. GPE was administrated by oral gavage for 8 weeks to mice fed an HFD. Total RNA was isolated from epididymal adipose tissues, reverse transcribed, and used for qRT-PCR. Relative mRNA expression levels of *Fas* (**A**), *Ap2* (**B**), *Cpt1* (**C**), and *Hsl* (**D**) in adipose tissues. Target mRNA expression was normalized to that of *Gapdh* and is presented relative to the CD group. Each bar represents the mean ± SD (*n* = 10). *** *P* < 0.001 significantly different from the CD group. Different letters indicate significant differences among the HFD, GPE 100, GPE 300, and Orlistat 30 groups at *P* < 0.05. CD: control diet; HFD: high-fat diet; GPE: *G. pentaphyllum* extract; GPE 100: HFD + 100 mg/kg body weight/day; GPE 300: HFD + 300 mg/kg body weight/day GPE; Orlistat 30: HFD + 30 mg/kg body weight/day Orlistat.

**Figure 4 nutrients-11-02475-f004:**
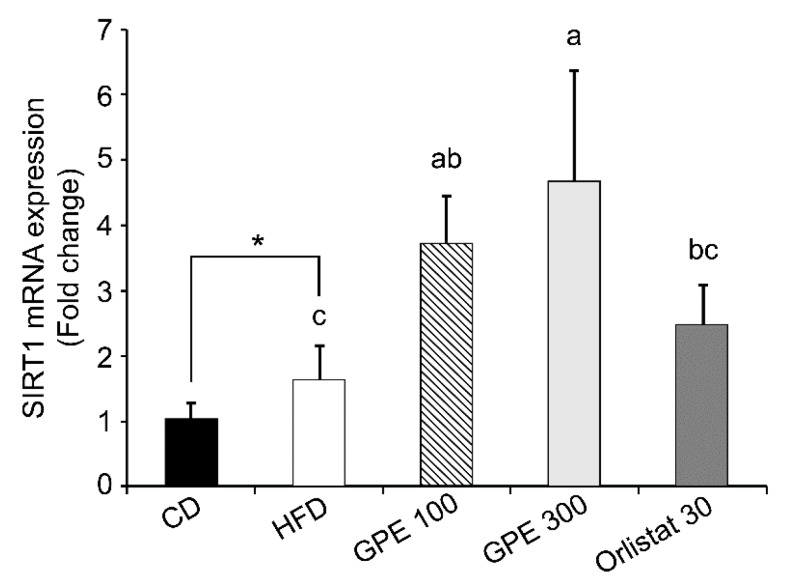
Effect of GPE treatment on the expression of *Sirt1* in the epididymal adipose tissues of HFD-fed C57BL/6N mice. GPE was administrated by oral gavage for 8 weeks to mice fed an HFD. Total RNA was isolated from epididymal adipose tissues, reverse transcribed, and used for RT-qPCR. Target mRNA expression was normalized to that of *Gapdh* and is presented relative to the CD group. Each bar represents the mean ± SD (*n* = 10). *** *P* < 0.001 significantly different from the CD group. Different letters indicate significant differences among the HFD, GPE 100, GPE 300, and Orlistat 30 groups at *P* < 0.05. CD: control diet; HFD: high-fat diet; GPE: *G. pentaphyllum* extract; GPE 100: HFD + 100 mg/kg body weight/day; GPE 300: HFD + 300 mg/kg body weight/day GPE; Orlistat 30: HFD + 30 mg/kg body weight/day Orlistat.

**Figure 5 nutrients-11-02475-f005:**
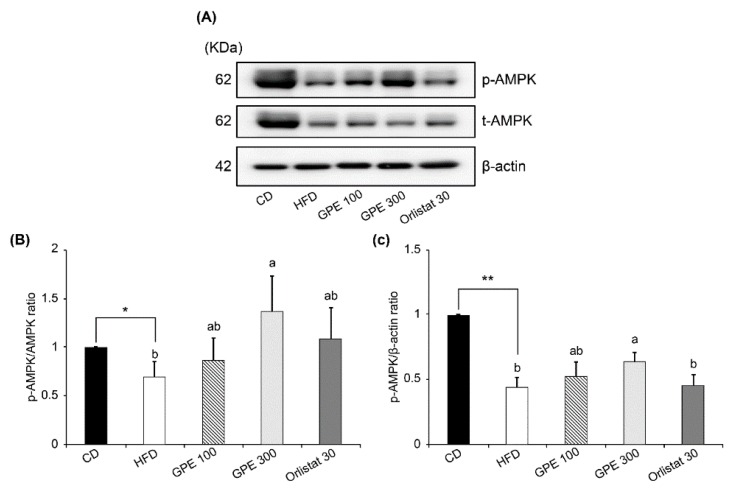
Effect of GPE treatment on the phosphorylation of AMPK in the epididymal adipose tissues of HFD-fed C57BL/6N mice. GPE was administrated by oral gavage for 8 weeks to mice fed an HFD. Total lysates of epididymal adipose tissues were prepared and analyzed by Western blotting with the indicated antibodies. (**A**) Photographs of Western blots representative of three independent experiments are shown. (**B**, **C**) Quantitative analysis of Western blot data. Protein abundances were normalized to β-actin and are expressed relative to CD levels. Each bar represents the mean ± SD (*n* = 5). *** *P* < 0.001 significantly different from the CD group. Different letters indicate significant differences among the HFD, GPE 100, GPE 300, and Orlistat 30 groups at *P* < 0.05. CD: control diet; HFD: high-fat diet; GPE: *G. pentaphyllum* extract; GPE 100: HFD + 100 mg/kg body weight/day; GPE 300: HFD + 300 mg/kg body weight/day GPE; Orlistat 30: HFD + 30 mg/kg body weight/day Orlistat.

**Table 1 nutrients-11-02475-t001:** Sequences of primers used in this study.

Target Gene	Forward Primer (5′-3′)	Reverse Primer (5′-3′)
*Ap2*	GGATTTGGTCACCATCCGGT	TTCACCTTCCTGTCGTCTGC
*Cebpa*	TGGACAAGAACAGCAACGAGTAC	TGGACAAGAACAGCAACGAGTAC
*Cpt1*	GTGCTGGAGGTGGCTTTGGT	TGCTTGACGGATGTGGTTCC
*Fas*	AGGGGTCGACCTGGTCCTCA	GCCATGCCCAGAGGGTGGTT
*Hsl*	CCGTTCCTGCAGACTCTCTC	CCACGCAACTCTGGGTCTAT
*Pparg*	CAAAACACCAGTGTGAATTA	ACCATGGTAATTTCTTGTGA
*Sirt1*	GCAACAGCATCTTGCCTGAT	GTGCTACTGGTCTCACTT
*Srebp1c*	CACTTCTGGAGACATCGCAAAC	ATGGTAGACAACAGCCGCATC
*Gapdh*	AGGTTGTCTCCTGCGACT	TGCTGTAGCCGTATTCATTGTCA

**Table 2 nutrients-11-02475-t002:** Effect of *Gynostemma pentaphyllum* extract (GPE) treatment on body weight, body composition, and food intake in high-fat diet (HFD)-fed C57BL/6N mice.

	CD	HFD	GPE 100	GPE 300	Orlistat 30
Initial body weight (g)	21.4 ± 0.8	21.5 ± 0.8	21.7 ± 0.9	21.5 ± 0.7	21.7 ± 0.7
Final body weight (g)	30.1 ± 1.7	45.6 ± 1.8 ***^,a^	39.8 ± 1.3 ^b^	37.3 ± 2.3 ^c^	41.0 ± 1.8 ^b^
Body weight gain (g)	8.7 ± 1.1	24.1 ± 1.3 ***^,a^	18.1 ± 1.4 ^b^	15.8 ± 2.2 ^c^	19.3 ± 1.9 ^b^
Lean mass percentage (%)	72.6 ± 5.3	56.6 ± 0.8 ***^,b^	57.9 ± 2.0 ^a,b^	59.2 ± 1.4 ^a^	58.2 ± 1.4 ^a^
Fat mass percentage (%)	27.4 ± 5.4	43.3 ± 0.8 ***^,a^	42.1 ± 2.0 ^a,b^	40.8 ± 1.3 ^b^	41.8 ± 1.4 ^b^
Food intake (g/day)	2.69 ± 0.04	2.33 ± 0.08 ***^,a^	2.11 ± 0.05 ^c^	2.07 ± 0.07 ^c^	2.17 ± 0.02 ^b^
Food efficiency ratio ^1^	0.058 ± 0.007	0.184 ± 0.010 ***^,a^	0.153 ± 0.011 ^b^	0.137 ± 0.020 ^c^	0.159 ± 0.016 ^b^

^1^ Food efficiency ratio = weight gain/food intake. Data are expressed as the mean ± SD (*n* = 10). *** *P* < 0.001 significantly different from the control diet (CD) group. Different letters indicate significant differences among the HFD, GPE 100, GPE 300, Orlistat 30 groups at *P* < 0.05.

**Table 3 nutrients-11-02475-t003:** Effect of GPE treatment on serum glucose and lipid levels and serum alanine aminotransferase (ALT) and aspartate aminotransferase (AST) activities in HFD-fed C57BL/6N mice.

	CD	HFD	GPE 100	GPE 300	Orlistat 30
Glucose (mg/dL)	101.2 ± 25.6	166.1 ± 40.7 ***^,^^a^	137.0 ± 17.1 ^b^	106.3 ± 13.5 ^c^	135.3 ± 18.1 ^b^
Triglyceride (mg/dL)	35.7 ± 4.6	71.9 ± 9.6 ***^,^^a^	49.2 ± 9.0 ^b^	50.4 ± 9.7 ^b^	57.9 ± 13.6 ^b^
Total cholesterol (mg/dL)	94.3 ± 16.5	126.5 ± 29.1 **^,^^a^	125.7 ± 19.8 ^a^	100.8 ± 15.4 ^b^	139.3 ± 27.9 ^a^
LDL-cholesterol (mg/dL)	26.2 ± 7.0	37.9 ± 6.6 **^,^^a^	28.6 ± 3.8 ^b^	23.2 ± 4.7 ^c^	28.7 ± 4.4 ^b^
HDL-cholesterol (mg/dL)	63.2 ± 11.0	88.6 ± 34.8 *	79.8 ± 25.7	79.2 ± 9.6	89.2 ± 23.9
ALT (U/L)	80.8 ± 24.6	275.7 ± 103.2 ***^,^^a^	127.3 ± 40.0 ^b^	105.1 ± 30.4 ^b^	91.3 ± 19.5 ^b^
AST (U/L)	150.0 ± 51.5	247.8 ± 77.3 **^,^^a^	184.9 ± 49.1 ^b^	175.5 ± 43.9 ^b^	154.6 ± 32.7 ^b^

Data are expressed as the mean ± SD (*n* = 10). * *P* < 0.05, ** *P* < 0.01, *** *P* < 0.001 significantly different from the CD group. Different letters indicate significant differences among the HFD, GPE 100, GPE 300, Orlistat 30 groups at *P* < 0.05.

**Table 4 nutrients-11-02475-t004:** Effect of GPE treatment on adipose tissue weights in HFD-fed C57BL/6N mice.

	CD	HFD	GPE 100	GPE 300	Orlistat 30
Epididymal fat (g)	0.99 ± 0.32	2.47 ± 0.30 ***^,^^a^	2.40 ± 0.24 ^a^	2.10 ± 0.24 ^b^	2.35 ± 0.22 ^a^
Retroperitoneal fat (g)	0.58 ± 0.16	1.59 ± 0.22 ***^,^^a^	1.27 ± 0.12 ^b^	1.08 ± 0.10 ^c^	1.28 ± 0.09 ^b^
Mesenteric fat (g)	0.69 ± 0.18	1.87 ± 0.26 ***^,^^a^	1.18 ± 0.07 ^b,c^	1.01 ± 0.21 ^c^	1.23 ± 0.15 ^b^
Inguinal fat (g)	0.33 ± 0.08	0.58 ± 0.08 ***^,^^a^	0.36 ± 0.16 ^b^	0.33 ± 0.09 ^b^	0.36 ± 0.15 ^b^

Data are expressed as the mean ± SD (*n* = 10). *** *P* < 0.001 significantly different from the CD group. Different letters indicate significant differences among the HFD, GPE 100, GPE 300, Orlistat 30 groups at *P* < 0.05.

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
