# Peer review of "Gynostemma Pentaphyllum Extract Ameliorates High-Fat Diet-Induced Obesity in C57BL/6N Mice by Upregulating SIRT1"

_nutrients, 2019, doi:10.3390/nu11102475_

Round 1

Reviewer 1 Report

This is an interesting study investigating the anti-obesity effects of Gynostemma pentaphyllum in a high fat diet model of C57BL/6. Several parameters related to the body mass index, food intake and cholesterol measurements are done to confirm the mode of action of this compound. In addition, some molecular data are generated (the majority of them are related to mRNA expression levels) are also quantified to explain how GPE mediate this effect by targeting these pathways.

Title: The title of the manuscript is too long: authors can change it to Gynostemma pentaphyllum ameliorates high-fat diet-induced obesity in C57BL/6 4. The details concerning the extracts can be explained one time in the abstract and include it as abbreviations in the manuscript, without repeating them.

Abstract: please do not keep redefining compounds especially peroxisome proliferator-activated receptor-gamma and try to include an abbreviation for each protein in this section.

Introduction: try to include the half survival time in the body of these compounds.

Materials and methods:

Why this study is not done in rats? please explain why authors used the C57BL/6 mice, what is the benefits of this model in term of obesity?

Maybe it is important to assess the androgen levels in male mice as well as the testosterone to make a clear conclusion that this compound modulate the sex related hormones.

If authors use female mice, do you think that the you have same pattern of results. Do you think that this compound is able to reduce obesity in female mice? How about estrogens and progesterone levels?

Can authors include the catalogue number related to the compounds that are used in the manuscript especially the primers, The Trizol kit for the isolation of total RNA, Syber Green kit, and the primers. Also, It should be crucial to indicate the catalogue number related the primary antibodies (for the protein of interest and the beta actin) and the secondary antibodies used in Western blotting.

Statistical analyses: all the results that are generated in the manuscript should be represented as Mean +/- Standard deviation (SD).

Results:

Authors should analyse and quantify the expression level of SIRT1 and present it with the Q-PCR data to make a definitive conclusion about the regulation manner of this target in mRNA and protein levels. Gene expression are not enough to generalize the mode of action of this target following the administration of this herbal compound.

It should be essential to quantify the ratio related to P-AMPK/B-actin and present it as (C) in this fig.5.

Discussion: the section that is related to the discussion is big, please try to reduce it by focusing on discussing your insights with previous study and please include the most recent publication related to this compound.

Reviewer 2 Report

the topic and the results presented in this manuscript are interesting and within the scope of your journal.

Therefore, I recommend that this manuscript can be accepted and will be published.

Author Response

Thank you for kind reviewing the manuscript.